# Decay-Accelerating Factor Creates an Organ-Protective Phenotype after Hemorrhage in Conscious Rats

**DOI:** 10.3390/ijms232113563

**Published:** 2022-11-05

**Authors:** Milomir O. Simovic, Michael J. Falabella, Tuan D. Le, Jurandir J. DalleLucca, Yansong Li

**Affiliations:** 1Department of Organ Function Support, US Army Institute of Surgical Research, JBSA Fort Sam Houston, San Antonio, TX 78234, USA; 2The Geneva Foundation, Tacoma, WA 98402, USA; 3Department of Trauma/Critical Care, Tri-Service Research Laboratory, JBSA Fort Sam Houston, San Antonio, TX 78234, USA; 4Armed Forces Radiobiological Research Institute, Bethesda, MD 20889, USA

**Keywords:** hemorrhagic shock, complement, DAF, fluid resuscitation, ischemia-reperfusion injury (IRI), hypotensive resuscitation

## Abstract

Preclinical and clinical studies have shown that traumatic hemorrhage (TH) induces early complement cascade activation, leading to inflammation-associated multiple-organ dysfunction syndrome (MODS). Several previous studies have demonstrated the beneficial effects of complement inhibition in anesthetized (unconscious) animal models of hemorrhage. Anesthetic agents profoundly affect the immune response, microcirculation response, and coagulation patterns and thereby may confound the TH research data acquired. However, no studies have addressed the effect of complement inhibition on inflammation-driven MODS in a conscious model of hemorrhage. This study investigated whether early administration of decay-accelerating factor (CD55/DAF, a complement C3/C5 inhibitor) alleviates hemorrhage-induced organ damage and how DAF modulates hemorrhage-induced organ damage. DAF was administered to unanesthetized male Sprague Dawley rats subjected to pressure-controlled hemorrhage followed by a prolonged (4 h) hypotensive resuscitation with or without lactated Ringer’s (LR). We assessed DAF effects on organ protection, tissue levels of complement synthesis and activation, T lymphocyte infiltration, fluid resuscitation requirements, and metabolic acidosis. Hemorrhage with (HR) or without (H) LR resuscitation resulted in significantly increased C3, C5a, and C5b-9 deposition in the lung and intestinal tissues. HR rats had significantly higher tissue levels of complement activation/deposition (particularly C5a and C5b-9 in the lung tissues), a higher but not significant amount of C3 and C5b-9 pulmonary microvascular deposition, and relatively severe injury in the lung and intestinal tissues compared to H rats. DAF treatment significantly reduced tissue C5b-9 formation and C3 deposition in the H or HR rats and decreased tissue levels of C5a and C3 mRNA in the HR rats. This treatment prevented the injury of these organs, improved metabolic acidosis, reduced fluid resuscitation requirements, and decreased T-cell infiltration in lung tissues. These findings suggest that DAF has the potential as an organ-protective adjuvant treatment for TH during prolonged damage control resuscitation.

## 1. Introduction

Trauma-induced hemorrhage (TH) is the leading cause of early mortality in trauma patients, often in the pre-hospital environment [1,2]. Hemorrhage is associated with 85% of potentially survivable death in recent war conflicts [3]. Those who survive bleeding are prone to systemic inflammatory responses, despite broadly approved medical and surgical treatments [4]. The resuscitation strategy involving low-volume permissive hypotension, where relatively lower fluid volume maintains central organ perfusion and biochemical homeostasis and reduces the possibility of rebleeding under some conditions provided therapeutic gains [5,6,7].

Due to hemorrhagic shock, decreased O_2_ delivery to cells results in lactate production at the expense of oxidative phosphorylation [8]. Hemorrhagic shock emanates from ischemia-reperfusion injury (IRI). IRI is a paradoxical phenomenon of multifactorial causation that triggers an innate immune inflammatory response [8,9,10]. Accumulating evidence implicates complement in the pathogenesis of IRI and hemorrhagic shock. Acute blood loss and tissue injury activate the complement cascade in humans [11], and the production of anaphylatoxin C3a and C5a and membrane attack complex (MAC or C5b-9) may have a damaging effect [12]. Any trauma induces local complement activation and, in the event of severe tissue injury, also leads to systemic activation and severe inflammation-associated cytokine storm [13,14]. Massive complement activation was observed in a porcine hemorrhagic shock even before resuscitation [15]. Complement inhibition in animal hemorrhage models showed advantageous effects [16,17,18,19]. Our previous studies have shown the favorable effects of pharmacological targeting of complement as evidenced by improved hemodynamics, decreased organ damage, modulation of inflammatory responses, reduced fluid requirements, and increased survival in rats and swine with traumatic hemorrhage [14,20,21]. Our clinical and animal studies also showed a direct correlation between complement activation and outcomes in the setting of traumatic hemorrhage [22,23].

According to the current Committee on Tactical Combat Casualty Care’s and Prolonged Field Care Working Group’s Guidelines, a permissive (hypotensive) resuscitation strategy is a fluid therapy recommendation for tactical combat TH casualty care at the point of injury or prehospital environments [24,25]. Hypotensive resuscitation is mainly examined and indicated for hemodynamically unstable patients typically presenting with penetrating trauma [26]. Permissive hypotension (restrictive fluid therapy) reduces blood loss, fluid and transfusion requirements, and severe postoperative coagulopathy. It may also cause prolonged IRI that aggravates complement C3/C5 activation in TH patients. This strategy is, however, contraindicated in patients with traumatic brain injury [27]. Few studies have examined the complement activation in animal models of hemorrhagic shock with hypotensive resuscitation [5,15,28,29]. In the setting of permissive hypotension, the complement inhibition improved hemodynamic parameters and responsiveness to fluid resuscitation in hemorrhaged rats [5] and improved survival and reduced fluid resuscitation requirements [28,29].

Modeling TH in *in vivo* animal models remains necessary for mechanistic and discovery studies. However, this also represents a challenge as, in contrast to human studies, performing animal TH generally requires anesthesia to restrain the animals and their gross body movement. Anesthetic agents profoundly affect the immune response, microcirculation response, and coagulation patterns [30,31] and thereby may confound the TH research data acquired. Hemorrhage in awake rats required approximately 25% more blood withdrawal than in anesthetized animals to achieve the same target mean arterial blood pressure. TNFα levels in blood serum and the target tissues of unanesthetized animals were significantly higher than in anesthetized rats [32]. No studies have addressed the effect of complement inhibition on inflammation-driven MODS in a conscious model of hemorrhage under hypotensive resuscitation.

Decay-accelerating factor (DAF) is a complement regulatory protein that inhibits complement C3/C5 activation by blocking the assembly of C3/C5 convertases, thus preventing the generation of C3a, C5a, and C5b-9 [33]. We hypothesized that early blocking of C3/C5 will ameliorate metabolic acidosis and reduce inflammation, organ damage, and fluid requirements. To test this hypothesis, we assess the effect of DAF on metabolic acidosis, fluid resuscitation requirements, and complement-driven MODS using a conscious rat model of hemorrhagic shock under the condition of hypotensive resuscitation.

## 2. Results

### 2.1. Effect of Recombinant Human DAF on Hemorrhage/Hypotensive Resuscitation-Induced Acidosis and Fluid Requirements

Early intravenous (IV) administration of recombinant human DAF (rhDAF) in rats with hemorrhage and lactated Ringer’s solution (LR)-resuscitation (HR) significantly reduced the concentration of blood lactate (*p <* 0.05, Figure 1A) and also markedly reduced the total volume of administered LR solution (*p <* 0.01, Figure 1B). Resuscitation with LR resulted in an elevation of the lactate level that was not associated with acidosis [34]. We still showed a difference in lactate levels between the HR and HR + DAF groups.

### 2.2. Effect of DAF on Hemorrhage/Hypotensive Resuscitation-Induced Acute Lung Injury (ALI)

We evaluated the effect of rhDAF on histopathological changes of the lung tissue after hemorrhage without LR-resuscitation (H) and HR. As shown in Figure 2A, the pulmonary tissue of H and HR rats showed distinct lung injury characterized by septal thickening, vascular congestion, disruption of the alveolar epithelium, and inflammatory cell infiltration (*p* < 0.01). Although the lung tissue from HR rats displayed additional septal hemorrhage and little more inflammatory cell infiltration than the H rats, there was no significant statistical difference between these groups, as it was indicated by semi-quantified total fluorescence (Figure 2B, *p* > 0.05).

rhDAF significantly reduced the H&E staining intensity of lung tissues in the HR, unlike in H rats (Figure 2B, *p* < 0.05). No significant statistical difference in the H&E staining intensity of the lung tissues between the H and HR rats treated with rhDAF (Figure 2B). rhDAF treatment exhibited a clear improvement of the morphological alterations either in H (Figure 2C, *p* < 0.05) or HR rats (Figure 2C, *p* < 0.01), but no significant difference in the injury score between hemorrhaged (H) and fluid-resuscitated hemorrhaged rats (HR) when both were treated with complement inhibitor (Figure 2C).

### 2.3. Effect of DAF on Hemorrhage/Hypotensive Resuscitation-Induced Pulmonary Complement Generation and Deposition

The lung tissue from the H rats and the HR rats showed an increased pulmonary deposition/generation of C3 and C5b-9 compared with the non-hemorrhagic rats (Sham, Figure 3). The majority of C5b-9 in the lung tissue co-localized with the vascular endothelium as shown by an increase in yellow fluorescent signal (Figure 3E). Interestingly, only partial co-localization of C3 and pulmonary endothelium was observed in HR rats, but not in H rats (Figure 3A). When compared with the H rats, LR-resuscitation displayed higher pulmonary deposition of C5a and C5b-9 (Figure 3C–F) and more C5b-9 pulmonary vascular deposition (Figure 3A,E). Little C5a pulmonary vascular deposition was observed in the H rats (Figure 3D). rhDAF administration significantly reduced the generation and deposition of C3 and C5b-9 (Figure 3) and decreased the pulmonary vascular deposition of C3 and C5b-9 (Figure 3C–F) compared with the H and HR rats. However, there was no significant difference in C3 (Figure 3B), C5a (Figure 3D), and C5b-9 deposition (Figure 3F) on the lung endothelium between H + DAF and HR + DAF groups.

### 2.4. Effect of DAF on Hemorrhage/Hypotensive Resuscitation-Induced Pulmonary C3 Synthesis and T lymphocyte Infiltration

The H rats treated with DAF and LR had significantly decreased pulmonary C3 mRNA expression (Figure 4) and pulmonary infiltration of T lymphocytes (Figure 5) compared with the HR group.

### 2.5. Effect of DAF on Hemorrhage/Hypotensive Resuscitation-Induced Acute Intestinal Injury

We also assessed histopathological alterations in the rat jejunum (Figure 6). The proximal jejunum section from a rat of the sham group demonstrated the normal architecture of the intestinal epithelium and wall. Conversely, histopathological changes of the proximal jejunum from the H/HR rats displayed remarkable intestinal damage depicted with epithelial cell sloughing, villi denuding, necrosis, and inflammation compared to the Sham (*p* < 0.01). However, the treatment with rhDAF attenuated the intestinal injury induced by resuscitated or non-resuscitated hemorrhage (*p* < 0.01).

### 2.6. Effect of DAF on Hemorrhage/Hypotensive Resuscitation-Induced Intestinal Complement Activation and Deposition

The complement C3 levels in the small intestinal tissues in the H/HR rats were significantly increased compared with the Sham group (*p* < 0.01; Figure 7A,B). The level of C3 in the gut tissues was higher in HR rats when compared with H rats, but the difference did not reach statistical significance (Figure 7B, *p* > 0.05). The increased level of C3 in the gut tissue of H/HR rats was significantly inhibited by the administration of rhDAF (*p* < 0.01). Next, we examined the effect of rhDAF on C5a deposition in the intestinal tissues using immunohistochemistry (Figure 7C,D). Micrographs from the tissues of injured rats with or without fluid resuscitation showed higher intensity of C5a than in the Sham group (Figure 7D, *p* < 0.01). An increased C5a deposition in the intestinal tissues of HR rats did not reach statistical significance when compared to H rats alone (Figure 7D, *p* > 0.05). In contrast, IV administration of rhDAF combined with LR significantly lowered the C5a levels compared with the HR group (HR vs. HR + DAF; *p* < 0.05), but rhDAF treatment in the H rats did not significantly decrease the generation of C5a when compared to the H group (H vs. H + DAF, *p* > 0.05). However, deposited C5a did not exhibit apparent colocalization with the intestinal microvasculature. Finally, we analyzed whether the administration of rhDAF affects the formation of the C5b-9 in the intestinal tissues of hemorrhaged rats with or without LR-resuscitation (Figure 7E,F). The C5b-9 level assessed by immunofluorescent staining with anti-rat C5b-9 was significantly higher in the intestinal tissues of H rats than in the Sham group (Figure 7E, *p* < 0.01). The C5b-9 level in the intestinal tissue was not significantly different after LR-resuscitation compared to those hemorrhaged only (Figure 7F; H vs. HR, *p* > 0.05), although there was a trend toward higher C5b-9 generation. The majority of C5b-9 deposition appeared along the intestinal vasculature. Intravenous administration of rhDAF significantly inhibited the deposition of C5b-9 (*p* < 0.01) and its colocalization with intestinal microvasculature in the H + DAF and HR + DAF rats.

### 2.7. Deposition of rhDAF in the Lung and Intestinal Tissues

We examined whether rhDAF administered IV is associated with rat tissues (Figure 8). The rat frozen lung and intestinal sections were stained with anti-human DAF antibody and probed with an appropriate secondary antibody labeled with Alexa Fluor 594. The fluorescent signal of DAF was not evident in the pulmonary and intestinal tissues in the rats that underwent Sham, H, or HR. On the other hand, deposition of DAF was noticeable in the lung and intestinal tissues of rhDAF-treated rats, either with hemorrhage or with bleeding, followed by resuscitation with LR.

## 3. Discussion

Previously, we and others have demonstrated the pathological role of complement activation in clinical trauma patients [23,35,36,37] and the beneficial effects of complement inhibition on MODS and mortality in preclinical anesthetized animal models of trauma [14,38], hemorrhage [20,21,29], and TH [6,23]. Using anesthetics in animal models often obscures crucial inflammatory, hemodynamic, and coagulation responses, thereby confounding pathophysiological results [4,5,39].

To minimize the anesthesia artifacts, in the present study, we investigated the effects of complement inhibition on inflammation-induced organ injury in unanesthetized rats with isobaric hemorrhagic shock under the condition of hypotensive LR resuscitation. The main findings were as follows: (1) early IV administration of rhDAF in rats subjected to hemorrhage effectively attenuated pulmonary and intestinal tissue damage; (2) rhDAF significantly improved metabolic acidosis and reduced fluid resuscitation requirements, and (3) the rhDAF mechanism of action appears associated with the blockage of complement-mediated vascular damage and inflammatory responses, as well as complement neosynthesis.

MODS represents a leading cause of late mortality following severe TH [40,41,42]. IRI and tissue damage after TH activates a complement cascade that plays a crucial role in inflammation-driven MODS [43]. Our previous findings showed early systemic/tissue complement activation associated with multiple-organ damage in rat and porcine models of TH [6,13,14,21,22,23,38]. Consistent with the previous findings, this study also demonstrated that increased generation and deposition of C3, C5a, and C5b-9 were associated with acute lung and intestinal injury, suggesting that early immunological damage control approaches aim to modulate TH-induced complement activation and thus may change TH management procedures to improve outcomes.

Hypotensive fluid resuscitation is one of the critical damage control resuscitation strategies for hemorrhagic shock in restoring intravascular volume and maintaining vital organ perfusion [25]. According to the current Committee on Tactical Combat Casualty Care and Prolonged Field Care Working Group’s Guidelines [23,24], a hypotensive resuscitation strategy is a fluid therapy recommendation for prehospital casualty care for TH patients. However, prolonged hypotensive resuscitation can aggravate TH-induced IRI, which may further activate the complement cascade [43]. Indeed, this study demonstrated that hemorrhaged rats that received LR resuscitation had significantly higher pulmonary levels of C5a and C5b-9 than hemorrhaged rats without LR resuscitation, which was associated with a trend toward higher lung injury. A report showed that fluid resuscitation including lactated Ringer’s solution in porcine liver injury and uncontrolled hemorrhage resulted in a significantly higher proinflammatory gene transcription in the lung tissue than in no resuscitation [44]. Therefore, to optimize TH damage control resuscitation, new strategies, such as immunological damage control resuscitative drugs targeting complement activation, are needed to address military prolonged field care and prolonged damage control resuscitation at the point of injury or prehospital environments.

DAF is a glycosyl phosphatidylinositol (GPI) anchored membrane protein found on erythrocytes, lymphocytes, granulocytes, endothelium, and epithelium. It inhibits the assembly of the classical and alternative pathway C3/C5-converting enzymes and C5b-9 generation [33,45]. Human DAF has a structure similar to rat DAF and shows cross-species reactivity [46]. The chosen dosage of rhDAF was in the titrated range used in our previous studies of hypoxia in rat primary neuronal cells [47], mouse ischemia-reperfusion [48,49], rat blast injury [14,38], and porcine hemorrhagic shock [20,29]. The time of DAF injection was based on our previous findings that early systemic complement activation after hemorrhage or blast injury is associated with the blood-brain-barrier (BBB) breakdown and multiple-organ damage [13,14,38,50]. In addition to the rhDAF association with the endothelium, it might directly enter the tissue via damaged vasculatures and/or post-injury infiltration of rhDAF-ligated circulating cells.

Our previous studies in anesthetized rats subjected to moderate blast overpressure illustrated that (1) rhDAF treatment (50 μg/kg) reduced brain and lung tissue complement deposition; (2) deposition of rhDAF was found in the brain and lung in rhDAF-treated rats with blast-induced neurotrauma and ALI; (3) deposition of rhDAF appeared to be associated with the cerebral and pulmonary endothelium; (4) early administration of rhDAF alleviated blast-induced neurotrauma and ALI and significantly reduced local and systemic inflammation; and (5) rhDAF treatment decreased the expression and colocalization of C3a and C3aR in the brain and lung tissues after blast injury [14,38]. We also revealed a dose-dependent rhDAF deposition in the lung and small intestinal tissues in anesthetized pigs subjected to isobaric hemorrhage and Hextend^®^ resuscitation. rhDAF treatment attenuated complement deposition (C3, C5, C5b-9) and activation in the lung and small intestinal tissues. Most C5b-9 in the small intestinal and pulmonary tissues was co-localized with the vascular endothelium. rhDAF treatment protected hemorrhage/resuscitation-affected multiple organs (such as lung, small intestine, liver, and kidney) from damage; reduced resuscitation fluid requirements, and improved survival. Even without fluid resuscitation, rhDAF treatment decreased tissue complement activation and deposition and reduced tissue damage in the same porcine isobaric hemorrhage [20,29]. Consistent with our previous findings, this study displayed several beneficial effects of rhDAF on reducing fluid requirements, improving metabolic acidosis, and ameliorating pulmonary and intestinal injury. The exact DAF mechanism of action on the beneficial effects is currently unknown.

Mast cells are extensively distributed in the connective tissue all around blood vessels and are involved early in inflammation. Mast cells and neutrophils activated by C3a and C5a cause vasodilation and extravasation of fluid [51] via the release of histamines [52]. A report showed that the complement activation products directly or indirectly alter vascular tone after trauma and hemorrhagic shock [53]. Porcine polytrauma, including controlled and uncontrolled hemorrhage, induced the early activation of the complement system, which was associated with increased cardiac interstitial inflammation [6]. Left ventricular expression of C3aR, C5aR1/2, and the changes in the deposition of C3a, C5, and C5b-9 early after trauma verified those cardiac pathohistological findings [6,43].

Mounting evidence suggests that trauma-induced complement activation may provoke endotheliopathy that leads to SIRS, vascular hyperpermeability, MOF, and thrombotic thrombocytopenic purpura via the “two-path unifying theory”: (1) C5b-9-endotheliopathy-cytokine storm and (2) C5b-9-endotheliopathy-platelet activation/endothelial exocytosis of uncommonly massive von Willebrand factor [54,55]. In line with these reports, this study demonstrated the colocalization of C5b-9/C3 and endothelial cells in the lung and intestine, suggesting that complement might also tightly regulate endotheliopathy. “Endotheliopathy of trauma” is a term first proposed by Holcomb and Pati [56], who attempted to describe a syndrome involving the early breakdown of the endothelial glycocalyx layer (EGL) after injury. Damage to the EGL increases vascular permeability and causes capillary leakage, resulting in the exposure of endothelial cells to circulating platelets and leukocytes. These events instigate the acute inflammatory response, which may alter coagulation [57,58]. The EGL regulates vascular permeability, mediates hemodynamic forces, including shear stress sensing, pressure, and circumferential stretch, and weakens blood cell–vessel wall interaction via mechanosensing and signal transduction [59]. As a result, systemic EGL breakdown triggers downstream responses, which leads to harmful systemic effects (i.e., thromboinflammation, edema, organ-barrier dysfunction, and endothelial dysfunction) [58,60,61].

In addition to microcirculatory effects, anaphylatoxin C5a and its interaction with C5aR1 further increase metabolic acidosis via C5a-C5aR1 signaling-mediated immunometabolic changes in neutrophils with enhanced glucose uptake and glycolytic flux and an elevated Na^+^/H^+^ exchanger type 1 activation, leading to a decrease in extracellular pH and an increase in intracellular pH [62].

Kulkarni et al. reported that human airway epithelial cells synthesize and store a large quantity of complement component C3, which may activate in response to stress [63]. Strainic et al. showed how locally generated complement fragments (C3a, C5a) deliver both costimulatory and survival signals to naïve CD4^+^ T cells [64]. This report may support our observation that DAF treatment reduced pulmonary infiltration of T lymphocytes in hemorrhaged rats. Anaphylatoxins, mostly C5a, take part directly in the activation of endothelial cells and leukocytes and indirectly by regulating TNF-α and IL-1 expression, which control adhesion molecules on both of these cells [65]. About 80% of DAF displays Brownian lateral motion inducing the inactivation of small amounts of deposited complement proteins on a cell membrane. Contrary to this effect, DAF shows relative immobilization in RBCs treated with rich complement deposition [66]. This relative DAF immobilization may characterize DAF monitoring on any cell membrane. The DAF treatment is less effective if it is administered approximately 30 min after the onset of traumatic insult [29], and this phenomenon may be explained by relative DAF immobilization owing to higher deposition of the complement activation products in an extended period after harmful stimuli.

Therefore, we speculate that the inhibition of complement activation by DAF limits local C3a/C5a generation, C3a/C5a interaction with their receptors (C3aR/C5aRs), and C5b-9 formation as well as C5b-9 accumulation on the cardiovascular endothelium, which could be playing crucial roles in maintaining homeostasis in cardiomyocyte function, vascular integrity, immunometabolism, and immunocoagulation.

There are some limitations to this study. We did not analyze systemic biomarkers of complement activation, inflammatory response, and MODS, which is a significant limitation in this current study. Further studies are needed to investigate these changes in preclinical animal TH models. Using only male animals is the only seeming limitation. Substantial evidence shows beneficial effects of estrogen for an immune-inflammatory response, multiple organ damage/failure, and mortality in terms of response to traumatic shock in preclinical and clinical studies [67,68,69,70,71]. Since the protective effects of estrogen would interfere with our research goals, we used male rats in our experimentations. Analyzing other tissues, including the liver and kidneys, would probably provide an insight into DAF effects. We have analyzed the lung tissue since it is involved in trauma-induced acute lung injury (TIALI) [72]. The intestinal tissue is crucial in developing multiple organ damage as an inducer of inflammatory mediators and a site of end-organ injury [73]. The liver generates most of the complement proteins, and we preferred to examine organ tissues on the periphery of the leading complement producer.

In summary, supplying tissue oxygenation with adequate fluid resuscitation is considered critical in the medical management of hemorrhagic patients. However, despite these reasonable therapeutic principles, the most effective resuscitation approaches often remain debatable. This situation is intrinsically related to the ischemia-reperfusion phenomenon. The data from this study suggest that the early inhibition of complement cascade may provide reliable control of illusive IRI conditions and fluid resuscitation requirements. Thus, targeting the complement may represent a new approach to immunological damage control resuscitation that may prevent MODS thereby improving survival in TH patients.

## 4. Materials and Methods

### 4.1. Animal Study

The study adhered to the principles stated in the Guide for the Care and Use of Laboratory Animals and was approved by the Walter Reed Army Institute of Research Institutional Animal Care and Use Committee (approval code: 11-OUMD; approval date: 13 January 2011) and performed in a facility accredited by the Association for Assessment and Accreditation of Laboratory Animal Care, International. This research was conducted in compliance with the Animal Welfare Act and other federal statutes and regulations related to animals and experiments involving animals.

### 4.2. Reagents

Recombinant human CD55/DAF was obtained from R&D systems (Minneapolis, MN, USA). Chicken anti-mouse C3/C3a, goat anti-chicken IgY (H&L), rabbit anti-rat T cell receptor (TCR), and mouse anti-rat endothelial cell antibodies from Abcam Inc. (Waltham, MA, USA) were used. Biotin rat anti-mouse C5a and streptavidin-fluorescent isothiocyanate conjugated antibodies were purchased from BD Pharmingen (San Jose, CA, USA). Mouse anti-rat C5b-9 primary monoclonal antibody was obtained from Hycult Biotech (Plymouth Meeting, PA, USA). Goat anti-human CD55 antibody were purchased from Santa Cruz Biotechnology Inc. (Santa Cruz, CA, USA). The goat anti-human C5 antibody was from Quidel (San Diego, CA, USA). Goat anti-chicken Alexa Fluor 488, goat anti-mouse Alexa Fluor 594, donkey anti-goat Alexa Fluor 647, goat anti-mouse biotin-Alex Fluo 488 IgG (H + L), streptavidin Alexa Fluor 488 conjugated secondary antibodies and ProLong Gold antifade reagent were from Invitrogen (Carlsbad, CA, USA).

### 4.3. Surgical Procedures

Male Sprague Dawley rats (380–480 g; Charles River Laboratories, Raleigh, NC, USA), 6-month-old, matured animals as analogs for adult people, underwent surgical catheter placement (RenaPulse™ RPT-0.040” × 0.025”, Braintree Scientific Inc., Braintree, MA, USA) under sterile conditions into the femoral artery and vein under isoflurane anesthesia (5% induction, 2% maintenance; Minrad International, Inc., Buffalo, NY, USA). Immediately after placement, the catheters were flushed with a total of 0.3 mL of 1% heparin solution to maintain catheter patency. There was no heparin administration during the experiment or observation period. The cannulas were tunneled subcutaneously, exteriorized, and secured at the posterior neck exit site via a flexible button cannula guide sutured to the skin. The catheters were then connected to the corresponding fluid reservoir and blood pressure monitor (BPA-400; Micro-Med, Louisville, KY, USA) through a two-channel fluid swivel. This arrangement allowed the animal free movement after recovery from the surgical preparation and prevented the animal from biting or twisting the cannulas, thus enabling the experiments to be conducted in awake rats. We did not provide the animals with preemptive analgesia.

### 4.4. Hemorrhagic Shock and Resuscitation

As previously described [5,74] and indicated in Figure 9, after surgical preparation, all animals recovered from anesthesia for a minimum of 1 h. At the time of experimentation, each enrolled animal was awake, alert, and without evidence of discomfort. No further anesthesia was maintained during the experimental protocol. The experimental design applied hypotensive resuscitation as a critical principle of the damage control resuscitation concept. The animals were enrolled in one of five experimental groups: hemorrhage (H; *n* = 6), H with resuscitation (HR; *n* = 6; lactated Ringer), H plus DAF (H + DAF, *n* = 7), HR plus DAF (HR + DAF; *n* = 7), and the sham group; the animals underwent identical anesthetic and surgical procedures but were not hemorrhaged or resuscitated (Sham; *n* = 5); blood samples (1 mL) from the enrolled animals were obtained at baseline and again 345 min later. The total number of animals used was 31. There were no dead or excluded animals during the experimentations. The hemorrhage model used in this study has been described and was previously shown to produce an approximate LD of 50% at 24 h [64]. The experimental hemorrhage and fluid resuscitation were conducted via the indwelling femoral arterial and venous catheter using a computer feedback and control program written in LabVIEW (National Instruments, Austin, TX, USA) to control a low-flow peristaltic pump (model P720; Instech Laboratories, Plymouth Meeting, PA, USA). Blood was withdrawn, and resuscitation fluids were given via the venous cannula. Shed blood and resuscitation fluids were held in separate reservoirs placed on a balance with fluid weights recorded every 5 s. Fluid volume was calculated from weight based on measured fluid density. MAP, shed blood volume, and intravenous fluid volume were continuously monitored and recorded every 5 s. Five-minute block averages of hemodynamic data were calculated and tabulated for analysis using Excel (Microsoft, Redmond, WA, USA). The experimental hemorrhage was initiated after a 20 min control period. Blood was withdrawn at a rate necessary to lower the MAP to 40 mmHg over 15 min, followed by withdrawal of additional blood as needed to maintain a MAP ceiling of 40 mmHg for 30 min. After the 30 min shock period, animals were randomized into treatment with rhDAF (60 µg/kg over 5 min, IV) or no treatment (an equal volume of normal saline administered over 5 min, IV). Animals then received an infusion of lactated Ringer’s solution as needed to raise the MAP to 60 mmHg over 10 min, followed by fluid infusion to maintain the MAP at a minimum of 60 mmHg for a period of 3 h. During this hypotensive support period, the fluid was infused only if the MAP decreased below 60 mmHg. No additional blood was withdrawn if the MAP rose above 60 mmHg. At the end of the hypotensive support period, the animals received an infusion of resuscitation fluid to raise the MAP to a target of 80 mmHg over 15 min, followed by infusion as needed to maintain a MAP at or above 80 mmHg for an additional 15 min. We euthanized the animals after this final resuscitation. To calculate MAP response per unit of infused fluid, we plotted the cumulative fluid infused during the resuscitation from the 60 mmHg threshold to the 80 mmHg threshold against the 5 min block-averaged MAP at the beginning and end of this resuscitation.

### 4.5. Tissue Harvest

The animals were euthanatized at the endpoint following the above procedures. Lungs and small intestines were removed, frozen in liquid nitrogen, and stored at −80 °C or fixed with 10% formalin/4% paraformaldehyde.

### 4.6. Histopathological Evaluation

As previously described [14,29], formalin-fixed lung and small intestinal tissues from individual animals were processed with paraffin. Sections were stained with hematoxylin and eosin (H&E) and observed, and histological images were recorded under a light microscope (Olympus Leica, AX80, Olympus, Center Valley, PA, USA) with a ×40/×20 objective.

Intestinal injury score: Mucosal injury of small intestines was graded on the six-tiered scale as described previously [14,29]. Briefly, the average of villi damage in an intestinal section (60–100 villi) was determined after grading each villus in the section on a 0–6 scale based on normal villus to damaged villus characterized by tip distortion, epithelial patchy disruption, epithelial cell sloughing, and denudation.

Lung injury score: For this purpose, 20 fields of H&E-stained lung sections from each animal were read at a magnification of ×400 [14,29]. The score given for each slide represented the mean score of these fields. Four parameters were examined: alveolar fibrin edema, alveolar hemorrhage, septal thickening, and intra-alveolar inflammatory cells. The changes were scored according to their extent (score 0, 1, 2, and 3 for an extent of 0%, <25, 25–50%, and >50%, respectively) and for the severity of the injury, (score 0 for no changes, score 1, 2, and 3 for more severe changes). The injury score represents the sum of the extent and the severity of the injury.

Measurement of H&E staining intensity of the lung tissues: Four to 6 images from each animal were analyzed using ImageJ software. Intensity measurements were assessed by setting a “threshold” using the thresholding tool of ImageJ [75].

### 4.7. Immunofluorescent Staining

This procedure was based on a method as described previously [14,29]. Briefly, paraformaldehyde-fixed lung and small intestine biopsies were snap-frozen to −70 °C, and sections were cut with a cryostat and fixed in cold methanol for 20 min. The fixed sections were permeabilized with 2% Triton X-100 in PBS for 10 min and blocked with 2% BSA in PBS for 30 min at room temperature (RT). The primary antibodies were incubated with the sections overnight at 4 °C and, after washing, the appropriate secondary antibodies labeled with FITC, Alexa Fluor 488, and 594 were incubated with the sections for 1 h at RT. After washing, the sections were mounted with ProLong Gold antifade solution containing 4′, 6′-diamidino-2-phenylindole (Invitrogen, Waltham, MA, USA), and visualized under a confocal laser scanning microscope (Radiance 2100; Bio-Rad, Hercules, CA, USA). Obtained digital images were processed, and complement fluorescent signals were quantified by the Adobe Photoshop software and ImageJ software. The number of stained T lymphocytes (TCR positive cells) and a total number of cells in 20 randomly selected fields at ×400 magnification of a given lung section were determined.

### 4.8. Total RNA Isolation and Reverse-Transcription Polymerase Chain Reaction

Total RNA was extracted from the lung tissues using an RNA isolation kit (RNeasy mini kit; Qiagen, Valencia, CA, USA) following the user manual. The purity and concentration of RNA were assessed by optical density ratio (OD260/OD280) and RNA gel electrophoresis. Reverse transcription (RT) with the total RNA was performed using an RT cDNA kit (Life Technologies, Grand Island, NY, USA). Expression quantification of C3 and β-actin was determined by RT-PCR analysis. Relative quantification of C3 mRNA was calculated by normalizing it to the endogenous reference β-actin.

### 4.9. Circulating Lactate Concentration

Blood lactate levels were analyzed using i-STAT CG4+ Cartridges (Abbott Point of Care Diagnostics, Princeton, NJ, USA).

### 4.10. Statistical Analysis

Continuous data were checked for normality using a combination of descriptive plots (i.e., histogram and Q-Q plots) and the Shapiro-Wilk test. Unless otherwise stated, normally distributed continuous data were presented as mean ± SEM and tested for the mean difference between study groups. We used an unpaired *t*-test with Welch’s correction for blood lactate, fluid requirements, C3 mRNA, and T lymphocyte infiltration and one-way ANOVA followed by Tukey’s post-hoc test for tissue injury and complement activation and deposition. Statistical significance was determined at the 2-sided *p* < 0.05. All statistical analyses were performed using GraphPad Prism 9.0 (GraphPad Software, San Diego, CA, USA).

## Figures and Tables

**Figure 1 ijms-23-13563-f001:**
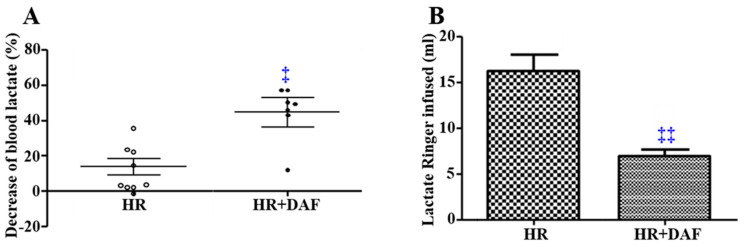
DAF improves metabolic acidosis and reduces fluid requirements in a rat model of hemorrhagic shock. Decrease in blood lactate (lactate level at 3 h post-LR resuscitation/lactate level at the end of shock ×100%). (**A**) Total volume of resuscitation fluid required ml. (**B**). ‡ *p* < 0.05 and ‡‡ *p* < 0.01 vs. HR (unpaired *t*-test with Welch’s correction).

**Figure 2 ijms-23-13563-f002:**
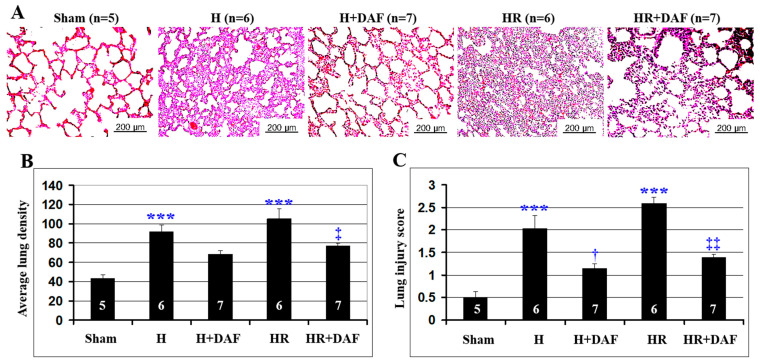
DAF attenuates hemorrhage-induced lung injury in rats. Histopathological changes of the lung on H&E slides. Representative photomicrographs are shown. (**A**) Quantification of the staining intensity of the lung tissue from the enrolled rats was performed using ImageJ software as described in Materials and Methods. (**B**) Cumulative histopathological lung injury scores for the sham, H, H + DAF, HR, and HR + DAF. (**C**) The numbers (5, 6, 7) in the figures mark the number of rats/tested samples per group. Original magnification ×400. Scale bar = 200μm. *** *p* < 0.001 vs. sham; † *p* < 0.05 vs. H; ‡ *p* < 0.05 and ‡‡ *p* < 0.01 vs. HR (one-way ANOVA followed by Tukey’s test).

**Figure 3 ijms-23-13563-f003:**
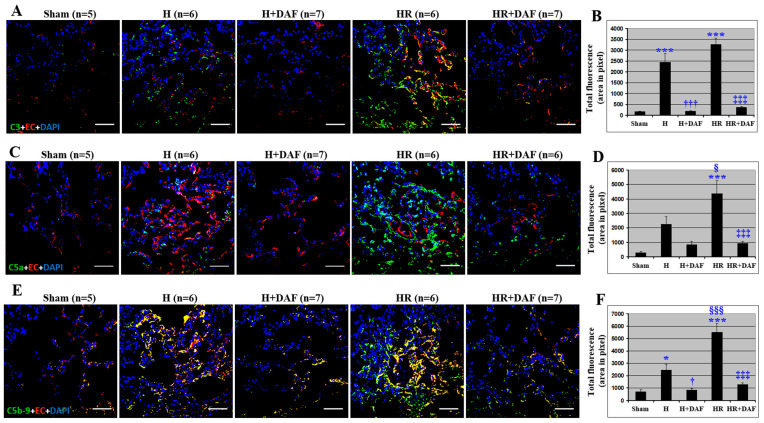
DAF inhibits deposition/generation of C3, C5a, and C5b-9 in rats subjected to hemorrhagic shock. C3 deposition. (**A**) C5a generation. (**C**), and C5b-9 formation. (**E**) on the lung endothelium were determined by immunohistochemistry using anti-C3, anti-C5a, anti-C5b-9, and anti-endothelial cell antibodies. Quantitative data of C3 (**B**), C5a (**D**), and C5b-9 (**F**) fluorescent signals of the lung tissues were measured by Photoshop and ImageJ software as stated in Materials and Methods. Original magnification ×400. Scale bar = 50 μm. * *p* < 0.05 and *** *p* < 0.001 vs. sham; † *p* < 0.05 and ††† *p* < 0.001 vs. H; ‡‡‡ *p* < 0.001 vs. HR; § *p* < 0.05 and §§§ *p* < 0.001 vs. H (one-way ANOVA followed by Tukey’s test).

**Figure 4 ijms-23-13563-f004:**
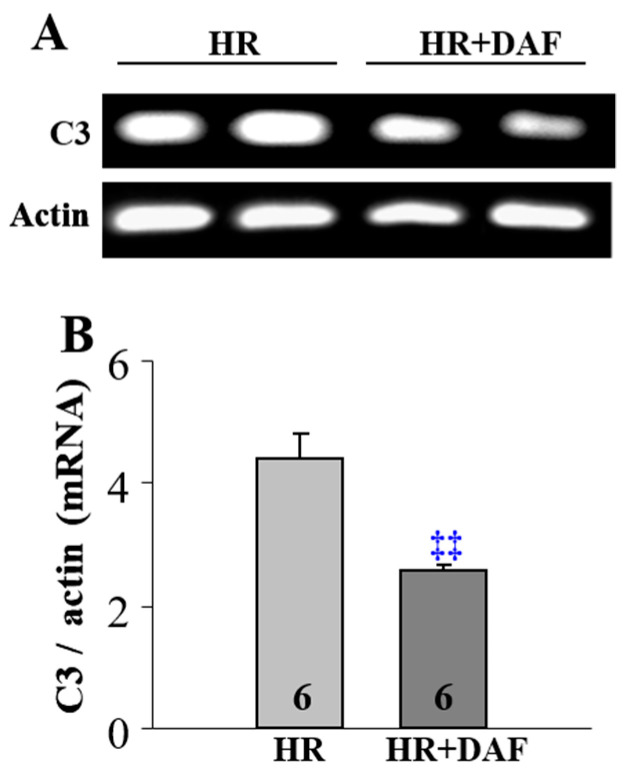
DAF treatment reduces pulmonary C3 mRNA synthesis in rats with hemorrhagic shock. Representative C3 mRNA expression (**A**) and quantitative expression of C3 mRNA (**B**) in the lungs. The number (6) in the figure marks the number of samples. ‡‡ *p* < 0.01 vs. HR (unpaired *t*-test with Welch’s correction).

**Figure 5 ijms-23-13563-f005:**
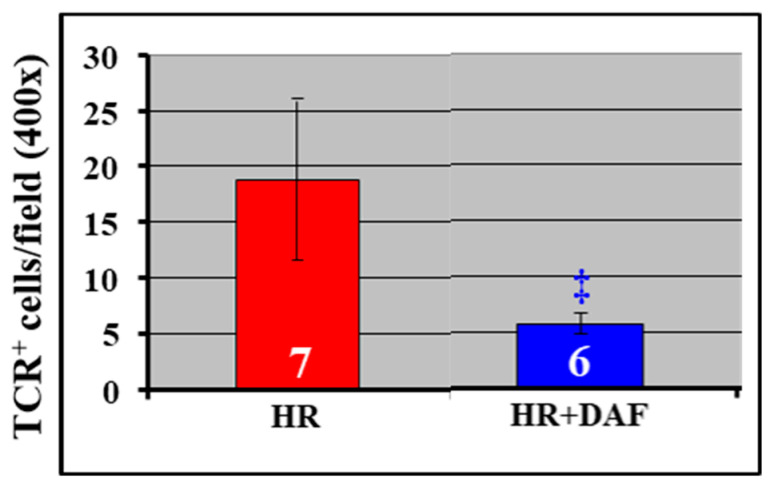
DAF treatment reduces pulmonary infiltration of T lymphocytes in rats with hemorrhagic shock. The number of positively stained cells with anti-TCR antibody and the total number of cells in 20 randomly selected fields at ×400 magnification of a given lung section were determined. The numbers (6, 7) in the figure mark the number of the samples. ‡ *p* < 0.05 vs. HR (unpaired *t*-test with Welch’s correction).

**Figure 6 ijms-23-13563-f006:**
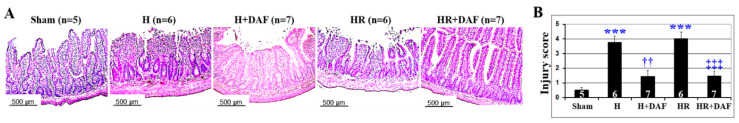
DAF mitigates hemorrhage-mediated small intestinal injury in rats. Representative histological alterations are presented. (**A**) Histological injury scores for the small intestine of rats. (**B**) The numbers (5, 6, 7) in the figure mark the number of samples. Original magnification ×200. Scale bar = 500 μm. *** *p* < 0.001 vs. sham; †† *p* < 0.01 vs. H; ‡‡‡ *p* < 0.001 vs. HR (one-way ANOVA followed by Tukey’s test).

**Figure 7 ijms-23-13563-f007:**
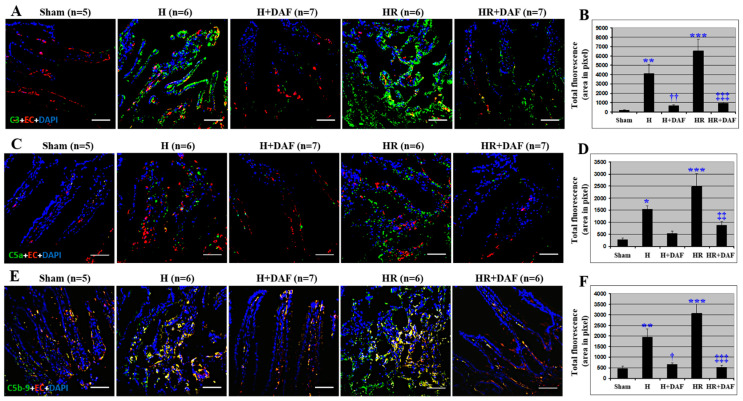
DAF inhibits deposition/generation of C3, C5a, and C5b-9 in the small intestine in rats subjected to hemorrhagic shock. C3 deposition (**A**), C5a generation (**C**), and C5b-9 formation (**E**) on intestinal endothelium were determined by histochemistry using anti-C3, anti-C5a, anti-C5b-9, and anti-endothelial cell antibodies. Quantitative data of C3 (**B**), C5a (**D**), and C5b-9 (**F**) fluorescent signals of the intestinal tissues were measured by Photoshop and ImageJ software as stated in Materials and Methods. Original magnification ×400. Scale bar = 50 μm. * *p* < 0.05, ** *p* < 0.01, and *** *p* < 0.001 vs. sham; † *p* < 0.05, and †† *p* < 0.01 vs. H; ‡‡ *p* < 0.01, and ‡‡‡ *p* < 0.001 vs. HR (one-way ANOVA followed by Tukey’s test).

**Figure 8 ijms-23-13563-f008:**
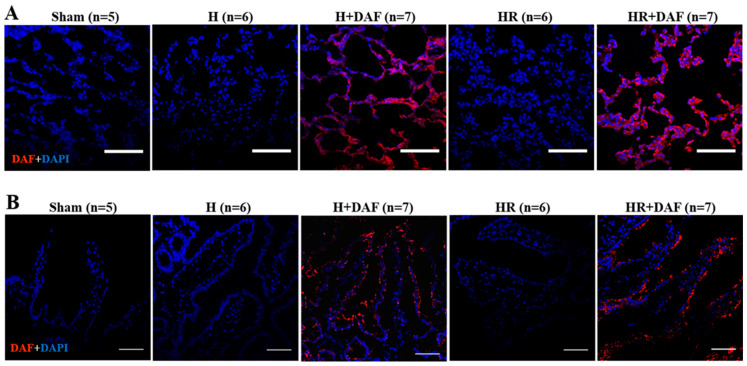
Intravenously administered DAF is deposited in the lung and intestinal tissues of rats. Representative lung (**A**) and intestinal (**B**) photomicrographs of immunofluorescent staining with anti-human DAF. Original magnification ×400. Scale bar = 50 μm.

**Figure 9 ijms-23-13563-f009:**
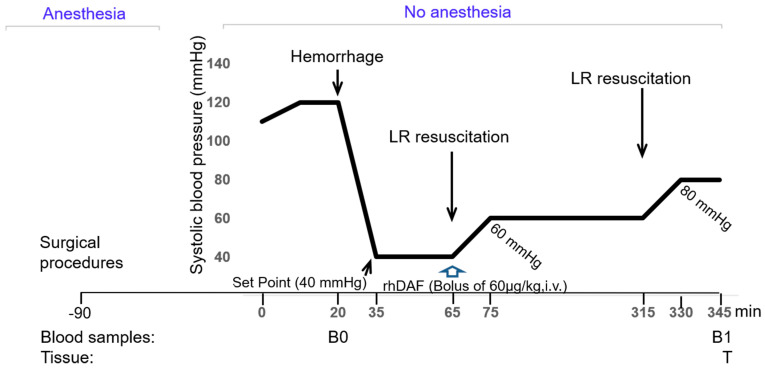
Scheme of the experimental design. Target mean arterial blood pressure (MAP), the blood, and tissue sample collection are shown. There was additional bleeding for maintaining the target pressure after the hemorrhage phase, but there was no shed blood infusion.

## Data Availability

Not applicable.

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
