# Peer review of "Decay-Accelerating Factor Creates an Organ-Protective Phenotype after Hemorrhage in Conscious Rats"

_ijms, 2022, doi:10.3390/ijms232113563_

Round 1

Reviewer 1 Report

The study to be assessed here is a sensibly conducted experiment with meaningful results concerning the use of DAF. However, the results are sometimes presented in a confusing, contradictory or even misleading manner, since it is not always clear whether the differences are significant. This should be better elaborated and not over-interpreted (as for the comparison between H and HR), since the study shows good results for the use of DAF. Please see the attachment. 

Author Response

General things:

  • As a limitation it should be explained why (only) male animals have been used and what this means for the results of the study. Also why only lung and intestine have been analyzed. How   about liver, kidney and so on?

Response: Thank you for the detailed review. Substantial evidence shows the beneficial effects of estrogen for an immune-inflammatory response, multiple organ damage/failure, and mortality in terms of response to traumatic shock in preclinical and clinical studies (Bosch et al., Mil Med Res. 2018; Yu et al., Shock. 2009; Doucet et al., PLoS One. 2010; Suzuki et al., J Cell Physiol. 2008; Marcolini et al., Anesthesiol Clin. 2019). Since the protective effects of estrogen would interfere with our research goals, we used male rats in our experimentations.

We agree that analyzing other tissues, including the liver, kidney, and others, would probably provide a further insight into DAF effects. We have analyzed the lung tissue since it is involved in trauma-induced acute lung injury (TIALI) (Fremont et al., 2010). The intestinal tissue is crucial in developing multiple organ damage as an inducer of inflammatory mediators and a site of end-organ injury (Hassoun et al., 2001). The liver generates most of the complement proteins, and we preferred to examine organ tissues on the periphery of the leading complement producer.

These explanations have been added to the manuscript      Line 300-308

  • Moreover, the authors should insert what age was used. Does this reflect the average trauma patient?
  • Rats age was 6 months. Matured animals as analogs for adult people were enrolled in experimentations.           Line 338-339
  • In the text it is often not clear if the authors report significant differences or only talk about tendencies as shown by different absolute values, that did not reach significance level. This should be changed in the whole manuscript. Actually, only significant results should be indicated as differences (see different sections) and others as trends as done in the explanation for Figure

Corrections regarding “significant differences” made in the whole manuscript.

For Immunhistological evaluation: What is the difference between generation and deposition and  pulmonary vascular deposition? How was this assessed? Either I did not totally understand, this is not clearly explained in the methods or the figure legends but the text points to a difference.

We used immunofluorescence analysis to detect complement factors. “Detection” may be the best expression to describe findings in immunofluorescence analysis. Although we wrote “generation/deposition,” we admit that such a description may be somewhat ambiguous. “Generation” refers to forming a factor at the spot where the element was found/detected. “Deposition” refers to already a formed factor somewhere else but deposited/laid down in the place where it is seen. We, however, cannot differentiate “formed/generated” from “deposited/laid down” in our data. 

When referred to the figures, it should be clear to which part (A, B, C,..). Sometimes explanations   in the text, the legend and the figure itself do not fit together. It has to be checked again by the authors that all parts tell the same story.

Corrections are made throughout the text, figures, and legends.

  • The size of the text in the figures is sometimes very small and difficult to read. This should be adjusted that the size is minimum

The size of text in the figures is adjusted.

  • This article mentions the effect of complement inhibitors on inflammation-induced organ damage in the discussion (lines 162, 168). However, analysis of changes in inflammatory cells (e.g. neutrophils, macrophages) in lung and small intestine tissues is lacking in the results of the article.
    We confined our interpretation of inflammation to pathohistological and immunofluorescent findings. As we wrote in the text, the lack of systemic biomarkers of complement activation, inflammatory response, and MODS is a limitation of this study. Line 297-298
  • In the mRNA results, we only see the effect of DAF on C3 transcription, is there a statistical effect on C5, C5-9b?

We did not analyze C5/C5b-9 mRNA because DAF’s mechanism of action is directly on C3 via preventing the formation of the C3-convertases.

Abstract: Are the named differences significant or only a tendency of the absolute values between the groups? When searched in the text, it seems that they are not (all) significant. Results in the abstract seem to show that no resuscitation is better but the results do not underline this for all indicated values - as we do not in all see significant changes between H and HR. In the Discussion, it is told correctly, that these differences are only significant for C5a and C5b-9 in the lung.

We removed this inconsistency from the whole manuscript.
Keywords: instead of, before hypotensive resuscitation

Graphical Abstract:

  • Abbreviation IRI should be

IRI is given and explained in Keywords                                Line 34

  • Some of the content is not discussed in the article, such as cell lysis, Inflammation.

We did not analyze systemic biomarkers of inflammation, but we discussed mast cells and neutrophils activated by complement factors.                                    Line 253-255

We centered our study on histopathology and fluorescent staining. Consequently, we rather discuss tissue damage instead of cell lysis and inflammation.

Introduction:

− Line 70 + 72: avoid beginning two sentences in a row with the same words “hypotensive resuscitation…”

 Permissive hypotension (restrictive fluid therapy) ….                  Line 75

− Line 76: It should be states with a few words what these studies found. Positive, negative, none effect?

Modified                                                                                         Line 78-82

− Line 77: The correct wording is “…in in vivo animal models…” One in is missing and in vivo should be written in italic.

Corrected                                                                                        Line 84

− Line 81: Are there studies on the comparison of TH in conscious and unconscious animals?

Revised                                                                                           Line 88-91

− Line 85: In the hypothesis, the authors propose that DAF will ameliorate metabolic acidosis. But there is no description of metabolic acidosis in vivo after hemorrhagic shock in the introduction.

Revised                                                                                   Line 56-57

Methods:

− No information about pain management is given. Was no analgesia given?

There was no analgesia                                                           Line 348-349

− The total number of animals used should be included. Did animals die or had to be excluded due    to some reasons?

31 rats                                                                                      Line 360-361

− There is no description of how to assess the “lung density” in the method.

Revised                                                                                    Line 428-430

− Statistics: Were data checked for normal distribution before analysis?

Yes. Continuous data were checked for normality using a combination of descriptive plots (i.e., histogram and Q-Q plots) and the Shapiro-Wilk test. 

We revised the Statistical analysis part accordingly.                  Lines 432-438
- Why were some data  analyzed with an unpaired T-test and others with a one-way ANOVA?

Because the unpaired t-test was used in this study to examine the effect of DAF on metabolic acidosis (i.e., blood lactate), fluid resuscitation requirements, and complements (i.e., C3/actin (mRNA) and TCR+) between the two specific groups of hemorrhage with lactated Ringer's resuscitation considered as a standard of care, HR vs. HR + DAF, as shown in Fig. 1, 4, and 5). One-way ANOVA was used to test for statistically significant differences between the means of these five study groups: Sham, H, H+DAF, HR, and HR + DAF.

- Which groups were compared with the T-test?

The HR with DAF group vs. the HR without DAF group.

Results:

− Fig 1: doubling of words in the legend should be removed

Revised                                                                                              

Fig 1. A, Lactate level at 3h post-LR resuscitation/lactate level at the end of shock x 100%.

− Figure 2:  A: space between H+DAF and (n=7) is missing and the background colors in A are not uniform.

Space made between H+DAF and (n=7) in Figures 2, 3, 6, and 7

  • B: Seems not to be described in the text.

Revised                                                                       Line 106-107

  • The abbreviations of the names of the groups need to be unified with those in the charts, such

as H and HS.

Revised throughout the whole manuscript

  • Magnification and scale bar can be put together, just like “Original magnification x400. Scale bar=200 μm.”

Corrected in all figures

  • The legend tells Unpaired t-test was used but in brackets also one-way ANOVA followed by Tukey’s test. What is correct? T-test is used for comparison between 2 groups and ANOVA for So ANOVA would be the correct test here for more than 2 groups.

We used one-way ANOVA

The text says: “As shown in Fig. 2, the pulmonary tissue of H and HR rats showed distinct lung injury characterized by septal thickening, vascular congestion, disruption of the alveolar epithelium, and inflammatory cell infiltration (p < 0.01).” No significance is shown in the figure for comparison between H and HR. Which part of Figure 2 (A-C) does this belong to? As we only see the injury score as the sum in C also no results can be seen for the single scoring points. Figure 2A only shows pictures without possible interpretation as significant or not.

Revised                                                                                               Line 117-123
No single scoring point just averaged lung injury scores

o Figure 3: A, C, E: space between H+DAF and (n=7) is missing.

Corrected

  • 3. What tests were used for analysis in B, D and F? Information is missing in the legend.

One-way ANOVA 
This information is added to the legend.                                           

Text is partly really small and difficult to read.

Text size is improved.

  • B, D, F: The lines showing significant differences are partly on the same level as the whiskers  and the bars instead of above the values. Sometimes the end in between two groups and not above one of them. This should be changed.

The position of the whiskers and bars is corrected.

  • The text says: “The lung tissue from the H rats and the HR rats showed an increased pulmonary deposition/generation of C3, C5a and C5b-9 compared with the non-hemorrhagic rats (Sham, 3)”. But no significant difference for C5a is shown in D between H and Sham. Only significant changes should be stated as differences.

There was no increase in C5a deposition                         Line 128 and 134

  • “When compared with the H rats, LR-resuscitation displayed higher pulmonary deposition of C5a and C5b-9 ( 3C-F), and more C3 and C5b-9 pulmonary vascular deposition (Fig. 3A and E)“. For C3 again no significant change is shown in the graph between H and HR.

“…more C3…” removed from the text. The graphs are corrected.          Line 132

  • How is the vascular deposition assessed?

The vascular deposition is defined by colocalization of C3 (yellow) and an endothelial marker (mouse anti-rat endothelial cell antibody). Line     129

  • “Little C5a pulmonary vascular deposition was observed in the H and HR rats ( 3 B).”
    Fig 3B shows C3 and not C5a.

It is C5a (Fig. 3D) only in the H rats                                                              Line     128-137

Figure 4:

− It is visible that the significance line is made of different drawings. This should be changed.

We corrected the issue.

Figure 5:

− Why is HR+DAF here on the left side but in all other figures on the right side of HR? This is a little confusing and should be changed.

The order of the groups in the figures is changed and made consistent.

− The vertical Whiskers of HR do not end at the horizontal line. There seems to be a mistake in

the graph.

We checked and corrected graphical issues.

− Unify the colors of the charts in Figure 4 and 5.

The colors of the charts are adjusted.

Figure 6:

− A: space between H+DAF and (n=7) is missing

The space is corrected.

− In B only a significant change between HR and Sham is shown, but not between H and Sham as stated in the text.

We corrected both issues.

− Unify the position, orientation and background of the small intestine tissue area shown in each picture, preferably the same as the sham group. The scale bar should not cover the tissue.

The position, orientation, and background of the small intestine tissue are unified, as suggested.

Figure 7:

− A, C, E: space between H+DAF and (n=7) is missing

Corrected

− B, D, F: The lines showing significant differences are partly on the same level as the whiskers and the bars instead of above the values. Sometimes the end in between two groups and not above one of them. This should be changed.

These graphical issues are improved.

− Lines 133 and 141: Immunohistochemical should be changed to Immunofluorescent

Revised                                                                                               Line 176         

− Line 137: Change DAF+HR to HR+DAF.

We changed the order of the groups as suggested.                                    Line 171

− Line 144: Change HS to H

Changed HS to H                                                                                Line 179

Line 152: HS should be changed to H as it is a little confusing for the reader.

Revised                                                                                               Line 187

Reviewer 2 Report

This paper uses a conscious rat model to investigate the hypothesis that early blocking of C3/C5 after trauma-induced hemorrhage will ameliorate metabolic acidosis, and reduce inflammation, organ damage, and fluid requirements. The hypothesis is not novel, since there are several previous studies showing that blocking complement proteins such as C1 or C5 improves outcomes following hemorrhagic shock (refs 5, 6). This same group has previously shown in a swine model that DAF treatment is protective against tissue injury following hemorragic shock (ref 19). The protocol and the analyses are very similar in both studies.  However, unlike the swine study which was performed under general anaesthesia, the current study is performed with conscious rats (to avoid confounding effects of anaesthesia).

Although the novelty is limited, the use of a conscious model, and a different animal species, is an important advance. The results of this study could assist progression of anti-complement treatments to clinical trials. Thus the results are worthy of publication.

Minor Concerns/Questions:

Technical:

1. I am concerned about the reliability of the C3 mRNA expression data, partly because of the choice of beta actin as a houskeeping gene, since beta actin levels are notoriously changeable in smooth muscle and fibrotic tissue (eg see http://genomebiology.com/2002/3/7/research/0034).  Also since C3 is primarily produced in the liver, what cells would be expressing C3 in the lung? Finally, the relevant issue is the level of complement activation in response to hemorrhagic shock - not so much the level of C3 which is a highly abundant plasma protein, but which would need to be synthesised after excessive blood loss.

2.  The authors state that there is a "lack of systemic biomarkers of complement activation".  However, there are ELISA kits available for complement pathway analysis, including for rat.

Clinical Implications:

Why was DAF chosen for these studies?

Is the response to DAF better than to anti-C5 mAb treatment?

Has C5aR1 inhibition been tested in this model?  This would provide information regarding the cause of tissue injury (C3b vs C5a vs C5b-9).

Would DAF be less expensive as a treatment than other drugs such as anti-C5 mAb or anti-C3 (compstatin/APL2/Pegcetacoplan)?

Although compstatin is only active against primate/human C3, so not able to be tested in non-primate animal models, it is already FDA approved for rare hematological disorders, and thus could be a future treatment option. Although targeting C3 (eg with DAF or compstatin) is more risky for chronic conditions (where immunosuppressive effects are a concern), this should not be a concern for one-off treatments such as for trauma-induced hemorrhage.

These questions should be discussed.

Author Response

I am concerned about the reliability of the C3 mRNA expression data, partly because of the choice of beta actin as a houskeeping gene, since beta actin levels are notoriously changeable in smooth muscle and fibrotic tissue (eg see http://genomebiology.com/2002/3/7/research/0034).  Also since C3 is primarily produced in the liver, what cells would be expressing C3 in the lung? Finally, the relevant issue is the level of complement activation in response to hemorrhagic shock - not so much the level of C3 which is a highly abundant plasma protein, but which would need to be synthesised after excessive blood loss.

Response: Thank you for your insightful review. Accumulating evidence demonstrates that the utility of the historically famous housekeeping genes (HKGs, e.g., β-actin, GAPDH, 18S rRNA) is limited in many cases due to their differential expression across species, tissue types, cell lines, developmental stages, and/or in response to experimental conditions/treatments (Chapman and Waldenström. PLoS One. 2015; Kozera and Rapacz. J Appl Genetics. 2013; Asiabi et al., J Assist Reprod Genet. 2020). In contrast to these reports, other research groups showed that β-actin is suitable/stable HKG for the gene expression studies for specific tissues/cells and pathological conditions (Röhn et al., Technol Cancer Res Treat. 2018; Tsotetsi et al., PLoS One. 2018; Day et al., Sci Rep. 2018; da Conceição et al., Acta Histochem. 2022). The selection of β-actin in this study as a reference gene (RG) for the normalization of pulmonary C3 mRNA expression is based on our previous study showing that pulmonary β-actin mRNA/protein exhibits low intergroup expression variability in a swine traumatic hemorrhage model (Campbell et al., Shock. 2016) and a rodent ischemia-reperfusion injury model (Lu et al., J Surg Res. 2011). We know that selecting/validating the proper HKGs is critical for gene expression studies of genes of interest (GOI). Concerning future RG selection, we will utilize two or more RGs and evaluate the expression stability of selected RGs. The precision and reproducibility of qPCR assays using the RefFinder web tool integrate data from the computational programs Normfinder, Bestkeeper, geNorm, and the comparative delta-CT methods.

Although the liver is the primary site for the synthesis of circulating complement proteins, emerging evidence reveals an extrahepatic synthesis of complement proteins in different tissues and cells. Various studies in both animals and humans have demonstrated that alveolar epithelial cells, airway epithelial cells, alveolar macrophages, and pulmonary fibroblasts synthesize several complement components (C3, C5, C6, C7, C8, C9, etc.,) under physiological and pathological conditions (Campbell et al., Shock. 2016; Pandya et al., Am J Respir Cell Mol Biol. 2014; Chaudhary et al., Mucosal Immunol. 2022; Strunk et al., J Clin Invest, 1988; Rothman et al., Am Rev Respir Dis. 1989; Varsano et al., Thorax. 2000; Hetland et al., Scand J Immunol. 1986), whereas pulmonary cells produce more complement proteins under disease conditions (Campbell et al., Shock. 2016; Pandya et al., Am J Respir Cell Mol Biol. 2014; Pettersen et al., Scand J Immunol. 1990; Bolger et al., Am J Physiol Lung Cell Mol Physiol. 2007). Furthermore, alveolar macrophage-derived serine proteinases cleaved locally synthesized complement proteins into active ones, which initiates pulmonary inflammation (Huber-Lang et al., Am J Pathol. 2002; Mol Immunol. 2001). This local production not only significantly contributes to the systemic pool of complement but also forms fully functioning complement pathways in the pulmonary niche, thereby participating in lung homeostasis and tissue damage (Campbell et al., Shock. 2016; Pandya et al., Am J Respir Cell Mol Biol. 2014; Laufer et al., Mol Immunol. 2001).

Complement factor 3 deficiency attenuated hemorrhagic shock-related hepatic injury and systemic inflammatory response syndrome after hemorrhagic shock and crushed thigh muscles (HS/T) in mice. HS/T resulted in C3 consumption in wild-type mice and C3 deposition in injured livers. C3-/-mice had significantly lower circulating IL-6, IL-10, and HMGB1 levels. Temporary C3 depletion by cobra venom factor preconditioning also led to reduced transaminases and a blunted cytokine release. C3-/- mice displayed well-preserved hepatic structure. The study provided conclusive data showing that complement activation was a central component of the initial response to HS/T (Cai et al., Am J Physiol Regul Integr Comp Physiol., 2010).

Next
Clinical Implications:

Why was DAF chosen for these studies?

Response: Since DAF affects C3/C5 activation, it enables us to follow the intensity of deposition/generation of C3, C5/C5a, and C5b-9 in two target tissues, the intestine, and lungs in the setting of hemorrhage and fluid resuscitation.

Is the response to DAF better than to anti-C5 mAb treatment?

Response: Yes. DAF treatment in this study demonstrated more beneficial effects (e.g., improved tissue damage and acidosis) than anti-C5 mAb treatment in a similar rat hemorrhagic model (Peckham et al., J Appl Physiol. 2007).

Has C5aR1 inhibition been tested in this model?  This would provide information regarding the cause of tissue injury (C3b vs C5a vs C5b-9).

Response: It is a great idea. We have not evaluated it in this model yet. But Harkin et al. showed that C5aR1 antagonist attenuates multiple organ injury in an unconscious rat model of hemorrhagic shock (Harkin et al., J Vasc Surg. 2004).

Would DAF be less expensive as a treatment than other drugs such as anti-C5 mAb or anti-C3 (compstatin/APL2/Pegcetacoplan)?

Tavneos (avacopan) is a C5a receptor (C5R1, CD88) inhibitor used to treat anti-neutrophil cytoplasmic autoantibody-associated vasculitis. Avacopan is indicated as an adjunctive treatment. It is used to treat vasculitis, such as granulomatosis with polyangiitis and microscopic polyangiitis. The lowest price of Tavneos (10 mg, 180 capsules) is around $14,200.

Paroxysmal nocturnal hemoglobinuria (PNH) is a rare complement-related hemolytic anemia with variable manifestations (Brodsky RA, Blood, 2021). EMPAVELI® (pegcetacoplan) is approved by the FDA for the treatment of PNH. The cost for Empaveli subcutaneous solution (1080 mg/20 mL) is around $37,107 for a supply of 160 milliliters. Eculizumab and ravulizumab are also the FDA-approved C5/terminal complement inhibitors.

We used DAF as a tool helping to understand the complement roles in traumatic hemorrhage. For possible use in trauma patients, we consider terminal complement inhibitors such as nomacopan (Yang et al., Br J Pharmacol., 2022).

Although compstatin is only active against primate/human C3, so not able to be tested in non-primate animal models, it is already FDA approved for rare hematological disorders, and thus could be a future treatment option. Although targeting C3 (eg with DAF or compstatin) is more risky for chronic conditions (where immunosuppressive effects are a concern), this should not be a concern for one-off treatments such as for trauma-induced hemorrhage.

Response: Compstatin, a cyclic tridecapeptide, probably would not be a concern for one-off treatments of trauma-induced hemorrhage. However, eliminating C3 from the cascade does not stop the lateral processes of the complement amplification and the terminal complement activation. Our recent study (Yang et al., Br J Pharmacol., 2022) supports the rationale for inhibiting the terminal complement pathway as a therapeutic strategy in traumatic hemorrhage.

These questions should be discussed.

We adjusted the text.
